PERSPECTIVE

# Unearthing the mechanisms of responsive neurostimulation for epilepsy

Vikram R. Rao [1✉] & John D. Rolston [2]

Responsive neurostimulation (RNS) is an effective therapy for people with drug-resistant focal epilepsy. In clinical trials, RNS therapy results in a meaningful reduction in median seizure frequency, but the response is highly variable across individuals, with many receiving minimal or no benefit. Understanding why this variability occurs will help improve use of RNS therapy. Here we advocate for a reexamination of the assumptions made about how RNS reduces seizures. This is now possible due to large patient cohorts having used this device, some long-term. Two foundational assumptions have been that the device's intracranial leads should target the seizure focus/foci directly, and that stimulation should be triggered only in response to detected epileptiform activity. Recent studies have called into question both hypotheses. Here, we discuss these exciting new studies and suggest future approaches to patient selection, lead placement, and device programming that could improve clinical outcomes.

Epilepsy is a common neurological disorder that afflicts one in 26 people during their lives[1]. People with epilepsy experience sporadic seizures, in which electrical activity is abnormal in the brain. These seizures usually start in the same parts of an individual's brain, described as the seizure foci. Despite many anti-seizure medications being available, one-third of people with epilepsy continue to have uncontrolled seizures[2]. For these individuals, described as having drug-resistant epilepsy, surgical removal (i.e., resection) of seizure-producing brain tissue offers the greatest chance of seizure control[3] (see Box 1 for a glossary of specialized terms used in this article). Benefits of resective surgery are immediate, and 50–80% of well-selected patients will be seizure-free[4]. Whilst generally safe and potentially curative, resective surgery has limitations. Removal of brain tissue can result in permanent neurological deficits[5]. Precise localization of the seizure focus is required when planning surgery to ensure the correct tissue is removed, but this is not always possible[6]. Effects of resection may not last long term[7,8]. Also, resection is typically not an option when seizures arise from brain regions involved in language or muscle control.

Responsive neurostimulation (RNS) is an alternative to resective surgery in which a device that is implanted into the skull delivers electrical stimulation through electrodes that are inserted directly into the seizure focus or foci[9,10]. RNS is designed to be a "closed-loop" therapy, which means the device continuously senses neural activity and delivers stimulation only in response to detection of particular patterns of brain activity that are known to precede the occurrence of seizures in a particular person[11]. RNS does not require removal of brain tissue and is reversible (i.e., the device can be removed). It can also be used when resection is not suitable, such as when there are bitemporal seizure foci (i.e., seizures arising from the temporal lobe on each side of the brain) or when there are concerns removal of brain tissue will be problematic[10]. Randomized controlled trials have established the efficacy of RNS for drug-resistant epilepsy involving one or two seizure foci, with patients experiencing an average reduction in seizure frequency of 75% after nine years of therapy[12].

---

[1] Department of Neurology and Weill Institute for Neurosciences, University of California, San Francisco, CA, USA. [2] Department of Neurosurgery, Brigham and Women's Hospital, Harvard Medical School, Boston, MA, USA. ✉email: vikram.rao@ucsf.edu

---

**Box 1 | Glossary**

**Closed-loop stimulation:** electrical stimulation of the brain that changes in response to ongoing biological signals. While these signals are usually direct recordings of neural activity, they can also be other physiological parameters, such as heart rate, or derived metrics, such as activity and rest periods.
**Deep brain stimulation (DBS):** direct electrical stimulation of deep brain structures, such as the thalamus or basal ganglia, to treat neurological and psychiatric disorders. In epilepsy, the thalamus is the most common target.
**Drug-resistant epilepsy:** seizures that are not controlled despite adequate trials of two or more anti-seizure medications.
**Electrocorticogram (ECoG):** a recording of brain activity from electrodes directly within the brain or on its surface.
**Epileptiform activity or discharges:** brief, paroxysmal electrical waveforms signifying brain irritability.
**Ictal**: relating to a seizure.
**Interictal**: between seizures.
**Open-loop stimulation:** electrical stimulation of the brain that is delivered in a constant or scheduled intermittent (on/off) manner, without regard to changes in brain activity.
**Resection or resective surgery:** surgical removal of pathological brain regions, which can include tumors or tissue generating seizures.
**Responder:** a person with ≥50% reduction in seizure frequency following treatment.
**Responsive neurostimulation (RNS):** electrical stimulation of the brain delivered in response to a detected event, like a seizure or burst of abnormal brain activity.
**Seizure focus or foci:** the brain area(s) from which seizures arise in a person with epilepsy.

---

However, RNS also has limitations. Patients seldom become seizure-free, thus RNS is often a palliative therapy where patients have to live with their disease rather than be cured. Maximal seizure reduction can take years to occur, with ongoing seizure-related morbidity and frequent clinic visits required for device tuning to optimize effectiveness. Although the long-term responder rate (i.e., the proportion of patients with ≥50% reduction in seizure frequency) is high (73%)[12], over a quarter of patients do not respond well to treatment. There are no established methods to determine which patients will benefit from RNS, which means that one in four patients currently undergo a costly, invasive surgical procedure and years of follow-up appointments with little ultimate benefit.

Fortunately, increased experience from users of RNS, advanced neuroimaging techniques, and analysis of data stored by the RNS device[13] are providing more information about how the device works. In this Perspective, we begin by outlining the current model that proposes how RNS reduces seizures. We then discuss recent studies that highlight limitations of this model, and we describe how this model should be revised to better describe the mechanism of action of RNS. We conclude by discussing the implications of this revised model for current clinical practice and how it can inform the design of next-generation neurostimulation devices.

## Current model for RNS seizure reduction
Nearly 70 years ago, pioneering work by Penfield and Jasper[14] showed that direct electrical stimulation of human cortex could attenuate spontaneous epileptiform discharges, laying the foundation for research into using neurostimulation to treat epilepsy. Decades later, evidence that stimulation of the seizure focus suppresses afterdischarges[15,16] informed development of an external RNS that involved a battery-operated desktop device[17]. RNS devices were further refined to develop a more compact internal RNS that was approved for clinical use in 2013 following clinical trials[18]. Thus, RNS was developed to be a "seizure stopper," similar to the rationale for the use of automated defibrillators during cardiac arrhythmias[19]. While the idea that seizures are aborted through acute, targeted electrical counter-stimulation of spiking occurring during seizures (ictal activity) is intuitive, limited experimental evidence supports it[20]. RNS stimulations are associated with acute reductions of spectral power[21] and with high-frequency desynchronization[22], effects reflecting decreased energy in brain waves. These effects could be expected to disrupt seizures, and, in cats, closed-loop stimulation of subcortical structures has been shown to suppress spiking

between seizures (interictal spiking) more effectively than random stimulation[23].

This mechanistic model for RNS assumes that RNS is most effective when stimulation is delivered as close as possible to the seizure focus and as early as possible after seizure onset. Thus, clinicians aim to localize seizures precisely (to inform RNS lead placement) and refine detection algorithms on the RNS device iteratively (to optimize sensitivity and specificity for seizures).

## Clinical reality following RNS
However, many of the clinical observations made during use of RNS are not compatible with the above model being the only mechanism by which RNS works. Application of the above model would predict that RNS should reduce the impact of seizures quickly, yet any improvement in outcome tends to be slow and steady over many years[12]. This contrasts with the impact of defibrillators, which immediately treat cardiac arrhythmias. RNS devices store records of brain waves as chronic electrocorticograms (ECoGs). Unequivocal examples of stimulation-induced seizure termination are uncommon in RNS ECoGs, even when accounting for preferential storage of long-duration epileptiform activity by the device[24], and, when present, tend not to be associated with clinical outcomes[25]. Most patients treated with RNS receive hundreds to thousands of brief stimulations each day, far exceeding the expected number of seizures; thus, most stimulation occurs in the interictal state, not during seizures. Although precise delineation of the seizure focus is thought to be required for RNS, patients with the most well-localized seizures (e.g., bilateral hippocampal sclerosis, which is typically treated by RNS) do not necessarily have the best outcomes, and treatment response in bitemporal epilepsy does not depend on whether stimulating electrodes are placed within or outside of the hippocampi[26]. Conversely, patients with poorly demarcated, spatially extensive (regional) neocortical seizure foci can do well with RNS even though the sizes of electrical stimulation fields near intracranial electrodes are smaller than the area of seizure foci[27]. Finally, long-duration, low-frequency RNS stimulation paradigms can be more effective than conventional short-duration, high-frequency ones, which suggests acute seizure disruption is not the only mechanism of action of RNS[28].

## Rationale for observed additional actions of RNS
Recent studies exploring structural and functional network connectivity within the brain, the timing of stimulation relative to dynamic brain states, and markers of neural plasticity provide

some explanation for these unexpected clinical observations. Although the RNS clinical trials were not powered for subgroup comparisons based on clinical or imaging features, there is emerging evidence that some brain networks may be intrinsically more responsive to RNS stimulation than others. Retrospective analysis of intracranial electroencephalography (EEG) in patients who were later treated with RNS revealed that ictal synchronizability, a metric reflecting the ease by which neural activity propagates through a functionally connected brain network, is inversely related to the degree of seizure reduction with RNS therapy[29]. Thus, RNS responders and non-responders can be distinguished prior to device implantation based on a biomarker derived from electrographic features of their seizures. Another recent study using pre-RNS magnetoencephalography (MEG) found that interictal global functional connectivity in certain frequency bands was lower in RNS non-responders compared with responders[30]. Taken together, it would seem the effectiveness of RNS therapy depends on intrinsic neurophysiological properties of seizures and the brain networks that give rise to them. For example, a speculative possibility is that interictal RNS stimulation can more readily diffuse through networks with high functional connectivity, which could potentiate its therapeutic effects, and that seizures less able to synchronize widespread networks are those most readily reduced by RNS stimulation.

If RNS efficacy depends on network characteristics, therapy could be expected to be more effective if the tissue activated by stimulating electrodes includes key nodes within these networks. Multiple studies of hippocampal neurostimulation have found no link between the precise anatomical location of electrodes and patient outcomes[26,31,32]. However, outcomes can be predicted when the specific brain circuit(s) being stimulated is known. In a study of RNS patients with leads extending into the hippocampus, seizure reduction was greatest when diffusion imaging revealed that the activated tissue was structurally connected to medial prefrontal cortex, anterior cingulate, and precuneus, nodes which tended to connect with more posterior regions of the hippocampus[31]. This study suggests that the current strategy of placing RNS leads based on anatomic landmarks should be revised to also include a consideration of patient-specific networks. These networks could potentially be delineated preoperatively and information about them used to target convergence points of white matter tracts implicated in the seizure. This approach is already being adopted when brain stimulation is used to treat movement disorders[33] and psychiatric disorders[34]. Another study that highlighted the importance of patient-specific functional connectivity for determining RNS efficacy used cortico-cortical evoked potentials to define "receiver" and "projection" nodes, areas of greater inward or outward connectivity, respectively, during intracranial EEG monitoring in patients who were later treated with RNS[35]. Clinical outcomes were significantly better when RNS electrodes were placed near receiver nodes. The findings suggest that epileptic networks may have points of vulnerability where targeted stimulation can exert high network controllability[36,37] and so might suppress seizures as well or better than stimulation directly at the seizure focus. Identification of these critical points, conceptually, the "Achilles' heels" of epileptic networks, could enable RNS leads to be better placed for optimal efficacy. This strategy is also in line with a larger body of evidence[38–40] that indicates that precise lead targeting within networks that involve thalamic nuclei is critical for the efficacy of deep brain stimulation (DBS), another treatment for epilepsy.

As the structural and functional network determinants of RNS efficacy are better understood, it is also becoming clear that temporal variables play a significant role. For a long time, seizures were thought to occur at random, but studies of RNS ECoG and other datasets have revealed the existence of a cyclical temporal structure in epilepsy[41]. This has led to the development of contemporary models of seizure timing which propose that there are alternating high and low states of seizure likelihood, which coincide with cycles in the rate of interictal epileptiform activity[42]. Cycles of interictal epileptiform activity exist over multiple timescales, from circadian to multidien (multi-day)[43–45], and seizures preferentially occur at certain phases of these cycles. Since cycles of epileptiform activity could be indicative of resting-state dynamics of the interictal network[46], the effects of RNS stimulation could depend on the specific network state at the time of stimulation. Indeed, a recent study found that the effects of changing RNS stimulation parameters (e.g., frequency, burst duration, and charge density) depend on the initial seizure risk state[47], with parameter changes effective at reducing seizures in one seizure risk state being less effective during another risk state. Consistent with this, a recent study examining how stimuli were distributed across these low- and high-risk states found improved outcomes when stimulations were delivered preferentially in low-risk states, i.e., those states less disrupted by ongoing epileptiform activity[48]. RNS stimulation parameters are typically adjusted every few months, remaining constant in the interim. Thus, observed outcomes may underestimate the potential impact of RNS therapy by being the compositive of potentially opposing effects during network state cycling[49].

The slow time course of seizure reduction with RNS therapy provides some of the strongest evidence for a long-term neuromodulatory effect on brain networks that generate seizures[50]. Instead of acute stimulation arresting seizures, chronic stimulation could render the network less prone to initiating seizures. This hypothesis has garnered recent support from clinical data. Analysis of stimulation effects on electrographic seizure patterns in RNS ECoGs showed that immediate inhibition of these patterns was not associated with clinical outcomes but that "indirect" effects, defined as those occurring before or at some latency (>10 s) from stimulations, were associated with clinical outcomes[25]. This suggests that the beneficial effects of stimulation may not be a consequence of direct involvement in seizures. Subsequently, analysis of chronic interictal RNS ECoGs revealed differential plasticity in functional network connectivity between RNS responders and non-responders[51]. Patients with the best outcomes from RNS are those with the greatest ability to reorganize functional network connectivity. Stimuli inducing this plasticity may be more effective when delivered during periods with less epileptiform activity[48], when endogenous neuroplasticity mechanisms are more active[52,53].

These effects may not be unique to RNS. An alternative neurostimulation therapy for epilepsy is DBS of the anterior thalamic nuclei. This uses scheduled intermittent stimulation (open-loop) to modulate functional network connectivity[54] and also takes years to maximally reduce seizures[55]. Chronic neuromodulatory effects could help explain the similarity in outcomes between open-loop neurostimulation modalities[55–57] and closed-loop RNS[12]. This also calls into question whether the effectiveness of RNS depends on its responsive, feedback activity. In principle, to determine whether stimulation at the start of seizures is necessary for seizure reduction, RNS detection settings could be tuned to patterns of neural activity that are not present at seizure onset, though this would necessarily compromise seizure quantification from the device[13].

## Emerging model for RNS seizure reduction

Multidimensional network determinants of RNS efficacy help explain observed variability in clinical outcomes. Ictal and interictal network connectivity, lead location in relation to key

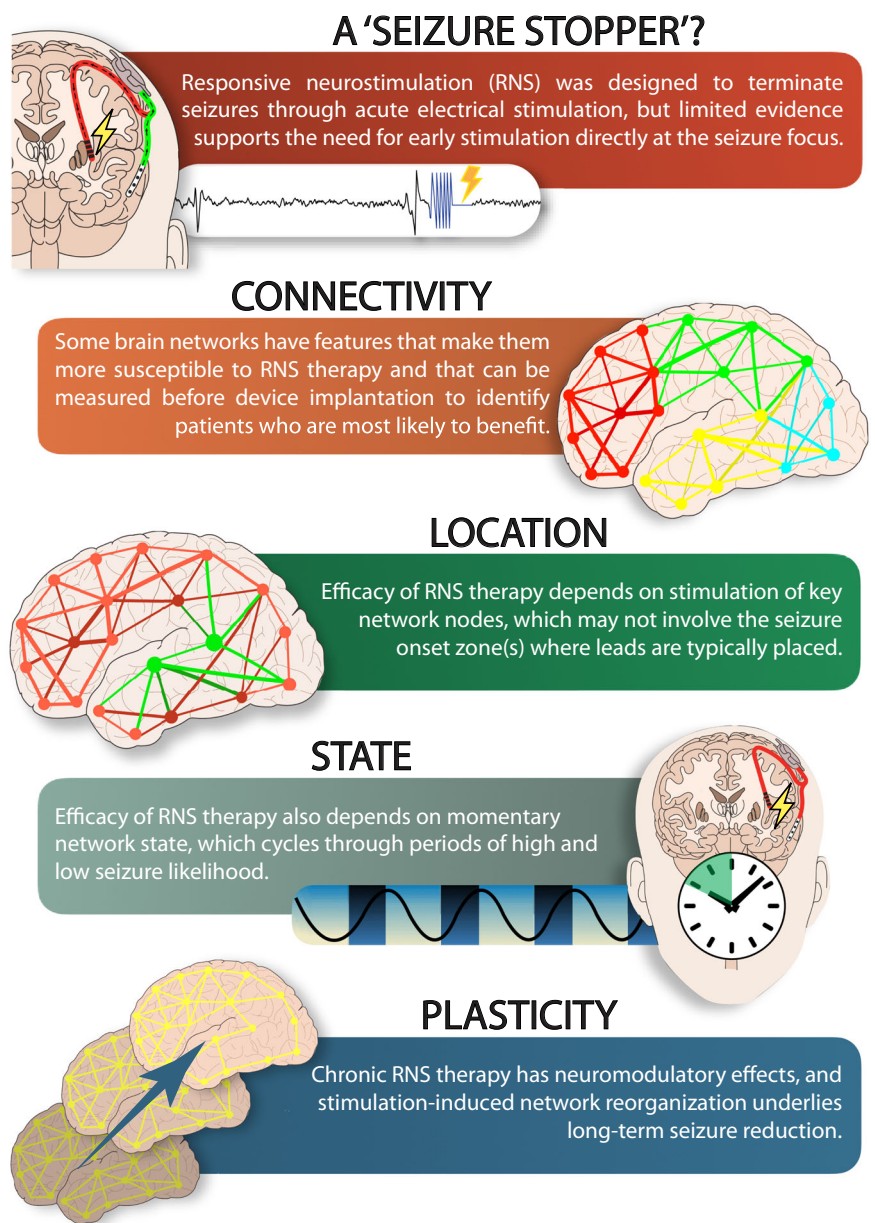

**Fig. 1 Infographic highlighting multidimensional network determinants of RNS efficacy.** Although RNS was originally conceived as a seizure stopper that works by acutely terminating seizures at their point(s) of origin, recent evidence reveals that this may not be its primary mechanism of action. Features of individual brain networks, including connectivity patterns, key structural and functional nodes, cyclical seizure risk states, and long-term plasticity collectively determine the extent of seizure reduction with RNS therapy.

structural and functional network nodes, cyclical network states, and long-term functional network reorganization collectively influence the likelihood that RNS will benefit a given individual (Fig. 1). Since current practice parameters do not generally consider these factors, we think it is remarkable that RNS works as well as it does. Hyperacute termination of ictal patterns may still play a role in seizure reduction with RNS, but chronic neuromodulatory effects are probably more important. Recent studies have revealed that one size does not fit all for epilepsy neurostimulation therapies, which need to be as diverse and dynamic as brain networks themselves. The implication of this new conceptual framework is that the clinical approach to virtually every step of RNS management, including patient selection, lead placement, and device programming, needs to be reconsidered. Some currently non-responding patients might benefit from the

consideration of dynamic network features when using RNS and other neurostimulation devices, such as thalamic DBS.

**Future directions**

Understanding of network-guided neuromodulation[50] might enable further personalization and better inform the choice and management of neurostimulation devices. Pre-surgical evaluations should focus on defining functional networks, in addition to identifying anatomical lesions and margins of seizure foci. Patient-specific models that use advanced neuroimaging techniques and integrate multi-dimensional network characteristics are already being employed[40] and are likely to become the standard of care. However, challenges remain that, if addressed, could accelerate clinical improvement. It is not yet known whether there are interventions that could catalyze network

reorganization, reducing the time required for patients to wait for meaningful seizure reduction. The appeal of palliative neuromodulation therapies will increase if seizure-free outcomes become similar to or exceed those currently attainable with resection. Indeed, as our ability to decode and manipulate brain networks increases, we might be able to help our patients' brains stop initiating seizures. Next-generation devices that possess enhanced capabilities will be essential to achieve this outcome. Such devices could benefit from a greater number of leads that interface with more brain networks and on-board artificial intelligence that enables real-time state analysis and adaptive stimulation[58]. We hope and anticipate that a future version of RNS may finally prove to be a seizure stopper.

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

## Acknowledgements

We thank Hargunbir Singh, MBBS for assistance with creating the infographic. V.R.R. is supported by an endowed professorship from the Ernest Gallo Foundation. J.D.R. is supported by the National Institutes of Health K23 (NS114178) and UH3 (NS109557) grants.

## Author contributions

V.R.R. and J.D.R. were both involved in the conception, writing, and approval of the manuscript.

## Competing interests

The authors declare the following competing interests: both authors are prior, but not current, consultants for NeuroPace, Inc., manufacturer of the RNS System, and are currently investigators in the NIH-funded RNS System Lennox-Gastaut Syndrome (LGS) Feasibility Study (ClinicalTrials.gov Identifier: NCT05339126). The authors declare no targeted compensation for this work.
