## [Peer review file · Communications Medicine]

Unearthing the mechanisms of responsive neurostimulation for epilepsyReviewers' comments:

Reviewer #1 (Remarks to the Author):

This manuscript nicely defines the current state of RNS treatment with an emphasis on the shortfalls and lack of response that has occurred in many patients. The authors describe the current seizure focus-centric approach embraced by most centers to define appropriate targets for RNS. They then succinctly define the shortcomings of this approach and further emphasize the long-term, neuromodulatory effects of the device as opposed to its "seizure-stopping" capabilities.

The authors describe some of the foundational work supporting a more network-centric interpretation of RNS outcomes from iEEG and MEG data. They then move to the matter of volume of tissue activation, again emphasizing the ascendancy of network over focus. The use of RNS trending data to identify and define circadian and multidien variations in seizure susceptibility has received appropriately increasing emphasis, but its application to timing of stimulation affords a unique tie-in with the network-centric bent of the manuscript. Finally, the idea that seizure-associated stimulation could be less effective over time than stimulation applied during more quiescent times raises questions regarding the benefits of closed versus open loop stimulation.

I do, however, have three questions for the authors.

- 1) The VTA study cited in lines 129-134 is quite interesting albeit in a limited subset of patients (hippocampal). It would be helpful to have more explanation as to how the authors foresee using personalized tractography and theoretical VTA data to define ideal targets for RNS.
- 2) Because the original manuscript is in press and not readily available for lines 136-139 (reference #30), the authors could better define how the original authors defined "receiver" and "projection" nodes as well as what was meant by "tended to be better" (i.e., was there a trend toward better outcomes versus a clearly significant difference)
- 3) It is expected that a greater proportion of the multidien cycle is spent in low-risk relative to high-risk states. Therefore, open-loop stimulation, which is impervious to the resting-state dynamics of the interictal epileptic network, should spend more time stimulating in low- than high-risk epochs of time. Since improved outcomes were noted when responsive stimulation was preferentially applied in low-risk states (reference #40, lines 156-159), one could speculate that open loop stimulation could have more neuromodulatory potential than closed loop stimulation although this is not borne out in the literature. I would be interested in the authors' thoughts regarding this conundrum.

It was a pleasure to read this manuscript and its description of the transition from the traditional notion of seizure foci – a notion still integral to the world of resective and ablative epilepsy surgery – to an emphasis on epileptic network dynamics – more applicable to the realm of neurostimulation and neuromodulation.

David Burdette

Reviewer #2 (Remarks to the Author):

This brief review examines the possible mechanisms underlying RNS, suggesting that some previously held idea are likely not correct.

Comments:

This brief review is well-written and provides a helpful perspective for those seeking to understand or explain how RNS may work. Both authors are experts in the field and between the two of them have written widely about RNS, DBS, and Neurostimulation in general for epilepsy.

Overall, this review might benefit from somewhat more context within the field of Neurostimulation, which would make the article much more interesting to readers. The abstract highlights two foundational assumptions that are questioned (lead location and closed loop

stimulation). However, the review itself does not seem to directly address the closed loop stimulation portion in comparison to open loop stimulation. For example, clinical results from DBS open loop devices are considered by most experts to be equivalent, and several studies exist comparing DBS with RNS (although retrospectively). In addition, there are studies dealing with open loop cortical stimulation for cortical stimulation (e.g Cukiert et al, 2017; Lundstrom et al, 2019) that seem to show similar results. How does this fit with the idea that stimulation must be responsive? It seems the closed loop aspect of RNS could readily be tested in RNS devices but never has been tested? In short, is it the case that the closed loop portion of the device has not been shown to be critical to its efficacy in any way?

Related to this, the assumption that lead location is not critical would ideally also be addressed in the context of what is known about DBS devices, where lead localization has been more extensively studied and seem to be important, if not critical.

Other comments:

- L62: responder rate may need to be defined for readers; similarly "meaningful improvement" is not clear
- L122-124 could benefit from an additional sentence to help clarify. It could be clearer how this fits with the prior logical flow of the review.
- L140: further clarity would be helpful
- L162: some of these issues have been recently addressed and reference could be helpful to Frauscher et al, Epilepsia 2023
- "Emerging Model" section would ideally contain clearer, distinct conclusions that summarize what was previously presented
- L205: "palliative therapies would be more palatable if they worked faster" – does DBS (with similar results) work faster?
- Perhaps an oversight related to competing interests? Both authors seem to have recently reported relevant disclosures.

Manuscript No.: COMMSMED-23-0533

Title: Unearthing the mechanisms of responsive neurostimulation for epilepsy

Corresponding Author: Vikram Rao

Response to Editor/Referee comments:

We appreciate the insightful, constructive reviews of our manuscript, and we are grateful for the opportunity to revise and resubmit it for consideration of publication in *Communications Medicine*.

Please find below our point-by-point responses to the Reviewers' concerns. The original concern is shown in *italics*. Our responses are shown in blue. All page numbers listed refer to pages on the manuscript version with tracked changes.

Reviewer: 1

This manuscript nicely defines the current state of RNS treatment with an emphasis on the shortfalls and lack of response that has occurred in many patients. The authors describe the current seizure focus-centric approach embraced by most centers to define appropriate targets for RNS. They then succinctly define the shortcomings of this approach and further emphasize the long-term, neuromodulatory effects of the device as opposed to its "seizure-stopping" capabilities.

The authors describe some of the foundational work supporting a more network-centric interpretation of RNS outcomes from iEEG and MEG data. They then move to the matter of volume of tissue activation, again emphasizing the ascendancy of network over focus. The use of RNS trending data to identify and define circadian and multidien variations in seizure susceptibility has received appropriately increasing emphasis, but its application to timing of stimulation affords a unique tie-in with the network-centric bent of the manuscript. Finally, the idea that seizure-associated stimulation could be less effective over time than stimulation applied during more quiescent times raises questions regarding the benefits of closed versus open loop stimulation.

I do, however, have three questions for the authors.

1) The VTA study cited in lines 129-134 is quite interesting albeit in a limited subset of patients (hippocampal). It would be helpful to have more explanation as to how the authors foresee using personalized tractography and theoretical VTA data to define ideal targets for RNS.

Response: Thank you for this comment. We envision, once these networks are validated in larger cohorts, defining these networks preoperatively and targeting these tracts directly, as is being done more frequently in DBS for movement disorders¹ and DBS for psychiatric disorders². We have now clarified this future direction in the text. [Page 8]

2) Because the original manuscript is in press and not readily available for lines 136-139 (reference #30), the authors could better define how the original authors defined "receiver" and "projection" nodes as well as what was meant by "tended to be better" (i.e., was there a trend toward better outcomes versus a clearly significant difference)

Response: We apologize for any confusion. We now define these terms more clearly, as follows:

“Another study highlighting the importance of patient-specific functional connectivity for determining RNS efficacy used cortico-cortical evoked potentials to define ‘receiver’ and ‘projection’ nodes—areas of greater inward or outward connectivity, respectively—during intracranial EEG monitoring in patients who were later treated with RNS³.” [Page 8]

We have also revised for clarity the statement on outcomes in this study: “Clinical outcomes were significantly better when RNS electrodes were placed near receiver nodes.” [Page 8]

If we are fortunate enough to have our manuscript accepted, and if the study by Kobayashi and colleagues is published by then, we will include a citation to the published work.

3) It is expected that a greater proportion of the multidien cycle is spent in low-risk relative to high-risk states. Therefore, open-loop stimulation, which is impervious to the resting-state dynamics of the interictal epileptic network, should spend more time stimulating in low- than high-risk epochs of time. Since improved outcomes were noted when responsive stimulation was preferentially applied in low-risk states (reference #40, lines 156-159), one could speculate that open loop stimulation could have more neuromodulatory potential than closed loop stimulation although this is not borne out in the literature. I would be interested in the authors’ thoughts regarding this conundrum.

Response: We appreciate the Reviewer’s reasoning here and fully agree that this is a conundrum in the field, one for which we do not have an answer, owing mostly to the lack of controlled studies that compare different neurostimulation modalities/strategies head-to-head. Since Reviewer 2 also invoked a comparison between open-loop and closed-loop stimulation, we have added text and references to highlight this point:

“Indeed, chronic neuromodulatory effects help explain the similarity in outcomes between open-loop neurostimulation modalities⁴⁻⁶ and closed-loop RNS⁷, and they call into question whether the effectiveness of RNS actually depends on its ‘responsive’ nature.” [Page 10]

It was a pleasure to read this manuscript and its description of the transition from the traditional notion of seizure foci – a notion still integral to the world of resective and ablative epilepsy surgery – to an emphasis on epileptic network dynamics – more applicable to the realm of neurostimulation and neuromodulation.

Response: We thank the Reviewer for these positive comments.

Reviewer: 2

This brief review is well-written and provides a helpful perspective for those seeking to understand or explain how RNS may work. Both authors are experts in the field and between the two of them have written widely about RNS, DBS, and Neurostimulation in general for epilepsy.

Overall, this review might benefit from somewhat more context within the field of Neurostimulation, which would make the article much more interesting to readers. The abstract highlights two foundational assumptions that are questioned (lead location and closed loop stimulation). However, the review itself does not seem to directly address the closed loop stimulation portion in comparison to open loop stimulation. For example, clinical results from DBS open loop devices are considered by most experts to be equivalent, and several studies exist comparing DBS with RNS (although retrospectively). In addition, there are studies dealing with open loop cortical stimulation for cortical stimulation (e.g Cukiert et al, 2017; Lundstrom et al, 2019) that

seem to show similar results. How does this fit with the idea that stimulation must be responsive? It seems the closed loop aspect of RNS could readily be tested in RNS devices but never has been tested? In short, is it the case that the closed loop portion of the device has not been shown to be critical to its efficacy in any way?

Response: The Reviewer is correct that the ‘responsive’ (closed-loop) nature of RNS, while appealing conceptually, has never been demonstrated to be critical for its efficacy. The closed-loop design of RNS stems from its original conception as a device for acute seizure termination, but, as we discuss at length, this is not likely to be its primary mode of action. The Reviewer is also quite right to point out that studies of open-loop neurostimulation modalities (we now cite the two studies suggested by the Reviewer, in addition to a study of long-term thalamic DBS outcomes by Salanova et al. (2021)) demonstrate outcomes that are similar to RNS, further casting doubt on the importance of closed-loop stimulation. However, it is not the case that the closed-loop aspect of RNS can be “readily” tested, because the device cannot be programmed to deliver stimulation in true open-loop fashion, thus precluding comparison of patients with open-loop vs. closed-loop RNS. In addition, head-to-head trials of different neurostimulation devices have not been done.

We have added the following text to incorporate these important points raised by the Reviewer:

“Indeed, chronic neuromodulatory effects help explain the similarity in outcomes between open-loop neurostimulation modalities⁴⁻⁶ and closed-loop RNS⁷, and they call into question whether the effectiveness of RNS actually depends on its ‘responsive’ nature.” [Page 10]

Related to this, the assumption that lead location is not critical would ideally also be addressed in the context of what is known about DBS devices, where lead localization has been more extensively studied and seem to be important, if not critical.

Response: We agree completely with the Reviewer that lead localization is critical. To harmonize recent studies on RNS lead targeting with the more extensive literature on DBS lead targeting, we have added the following text:

“This strategy dovetails with a larger body of evidence⁸⁻¹⁰ indicating that precise lead targeting in relation to networks that involve thalamic nuclei is critical for the efficacy of deep brain stimulation (DBS) in epilepsy.” [Page 8]

Other comments:

- L62: responder rate may need to be defined for readers; similarly “meaningful improvement” is not clear

Response: We now define responder rate as “the proportion of patients with $\geq 50\%$ reduction in seizure frequency”. We have also rephrased the second part of the sentence as: “...over a quarter of patients are not responders and may not experience worthwhile improvement with RNS.” [Page 4]

- L122-124 could benefit from an additional sentence to help clarify. It could be clearer how this fits with the prior logical flow of the review.

Response: We have added text to improve clarity and logical flow of this section:

“... the effectiveness of RNS therapy seems to depend on intrinsic neurophysiological properties of seizures and the brain networks that give rise to them. For example, a speculative possibility is that interictal RNS stimulation

can more readily diffuse through networks with high functional connectivity, which may potentiate its therapeutic effects, and that seizures less able to synchronize widespread networks are those most readily quelled by RNS stimulation.” [Page 7]

- L140: *further clarity would be helpful*

Response: We have added text to clarify this concept:

“... epileptic networks may have points of vulnerability where focal stimulation can exert high network controllability^{11,12} and, conceivably, can suppress seizures as well or better than stimulation directly at the seizure onset zone. Identification of these critical points—conceptually, the ‘Achilles’ heels’ of epileptic networks—could inform RNS lead placement for optimal efficacy.” [Page 8]

- L162: *some of these issues have been recently addressed and reference could be helpful to Frauscher et al, Epilepsia 2023*

Response: We thank the Reviewer for this suggestion, and we now cite this excellent work by Frauscher and colleagues.

“Thus, observed outcomes may underestimate the potential of RNS therapy by reflecting the net of potentially opposing effects during network state cycling¹³.” [Page 9]

- *“Emerging Model” section would ideally contain clearer, distinct conclusions that summarize what was previously presented*

Response: Our hope was the infographic Figure would serve to distill key conclusions from the studies that were reviewed. We have made some aesthetic improvements to this Figure for greater clarity and visual appeal. In addition, we have added the following sentence to this section to try and capture the essential sentiment of this perspective piece:

“Recent studies have revealed that one size does not fit all for epilepsy neurostimulation therapies, which need to be as diverse and dynamic as brain networks themselves.” [Page 11]

- L205: *“palliative therapies would be more palatable if they worked faster” – does DBS (with similar results) work faster?*

Response: We mention that RNS and DBS both take years to maximally reduce seizures, and clinical trial publications cited in the manuscript suggest comparable timecourses for improvement. We are not aware of data showing that DBS works more quickly than RNS.

- *Perhaps an oversight related to competing interests? Both authors seem to have recently reported relevant disclosures.*

Response: We have added a Competing Interests Statement to declare our prior roles as consultants for NeuroPace, Inc. and our ongoing involvement as investigators in the NIH-funded RNS-LGS Feasibility Study (NCT05339126). [Page 12]

References

- 1 Middlebrooks, E. H. *et al.* Neuroimaging Advances in Deep Brain Stimulation: Review of Indications, Anatomy, and Brain Connectomics. *AJNR Am J Neuroradiol* **41**, 1558-1568, doi:10.3174/ajnr.A6693 (2020).
- 2 Riva-Posse, P. *et al.* A connectomic approach for subcallosal cingulate deep brain stimulation surgery: prospective targeting in treatment-resistant depression. *Mol Psychiatry*, doi:10.1038/mp.2017.59 (2017).
- 3 Kobayashi, K. *et al.* Effective connectivity relates seizure outcome to electrode placement in responsive neurostimulation. *Brain Commun* **in press** (2023).
- 4 Cukiert, A., Cukiert, C. M., Burattini, J. A., Mariani, P. P. & Bezerra, D. F. Seizure outcome after hippocampal deep brain stimulation in patients with refractory temporal lobe epilepsy: A prospective, controlled, randomized, double-blind study. *Epilepsia* **58**, 1728-1733, doi:10.1111/epi.13860 (2017).
- 5 Lundstrom, B. N., Gompel, J. V., Khadjevand, F., Worrell, G. & Stead, M. Chronic subthreshold cortical stimulation and stimulation-related EEG biomarkers for focal epilepsy. *Brain Commun* **1**, fcz010, doi:10.1093/braincomms/fcz010 (2019).
- 6 Salanova, V. *et al.* The SANTÉ study at 10 years of follow-up: Effectiveness, safety, and sudden unexpected death in epilepsy. *Epilepsia* **62**, 1306-1317, doi:10.1111/epi.16895 (2021).
- 7 Nair, D. R. *et al.* Nine-year prospective efficacy and safety of brain-responsive neurostimulation for focal epilepsy. *Neurology* **95**, e1244-e1256, doi:10.1212/WNL.000000000010154 (2020).
- 8 Gross, R. E., Fisher, R. S., Sperling, M. R., Giftakis, J. E. & Stypulkowski, P. H. Analysis of Deep Brain Stimulation Lead Targeting in the Stimulation of Anterior Nucleus of the Thalamus for Epilepsy Clinical Trial. *Neurosurgery* **89**, 406-412, doi:10.1093/neuros/nyab186 (2021).
- 9 Schaper, F. *et al.* Mapping Lesion-Related Epilepsy to a Human Brain Network. *JAMA Neurol*, doi:10.1001/jamaneurol.2023.1988 (2023).
- 10 Warren, A. E. L. *et al.* The Optimal Target and Connectivity for Deep Brain Stimulation in Lennox-Gastaut Syndrome. *Ann Neurol* **92**, 61-74, doi:10.1002/ana.26368 (2022).
- 11 Khambhati, A. N. *et al.* Functional control of electrophysiological network architecture using direct neurostimulation in humans. *Netw Neurosci* **3**, 848-877, doi:10.1162/netn_a_00089 (2019).
- 12 Scheid, B. H. *et al.* Time-evolving controllability of effective connectivity networks during seizure progression. *Proc Natl Acad Sci U S A* **118**, doi:10.1073/pnas.2006436118 (2021).
- 13 Frauscher, B. *et al.* Stimulation to probe, excite, and inhibit the epileptic brain. *Epilepsia*, doi:10.1111/epi.17640 (2023).

Reviewers' comments:

Reviewer #1 (Remarks to the Author):

This manuscript nicely defines the current state of RNS treatment with an emphasis on the shortfalls and lack of response that has occurred in many patients. The authors describe the current seizure focus-centric approach embraced by most centers to define appropriate targets for RNS. They then succinctly define the shortcomings of this approach and further emphasize the long-term, neuromodulatory effects of the device as opposed to its "seizure-stopping" capabilities. The authors describe some of the foundational work supporting a more network-centric interpretation of RNS outcomes from iEEG and MEG data. They then move to the matter of volume of tissue activation, again emphasizing the ascendancy of network over focus. The use of RNS trending data to identify and define circadian and multidien variations in seizure susceptibility has received appropriately increasing emphasis, but its application to timing of stimulation affords a unique tie-in with the network-centric bent of the manuscript. Finally, the idea that seizure-associated stimulation could be less effective over time than stimulation applied during more quiescent times raises questions regarding the benefits of closed versus open loop stimulation. It was a pleasure to read this manuscript and its description of the transition from the traditional notion of seizure foci – a notion still integral to the world of resective and ablative epilepsy surgery – to an emphasis on epileptic network dynamics – more applicable to the realm of neurostimulation and neuromodulation. The authors have addressed this reviewer's questions, and from my perspective, this manuscript is ready for publication.

David Burdette

Reviewer #2 (Remarks to the Author):

Thank you for the changes.

One minor point:

Then authors may not have fully considered the initial comment:

"It seems the closed loop aspect of RNS could readily be tested in RNS devices but never has been tested? "

Would it not be readily accomplished to set RNS detectors randomly or tuned to something other than seizure onsets, thereby testing the closed loop component? To my knowledge this has never been attempted, but would in a scientific setting be one of the first experiments/tests performed.

The authors responded that "true open loop" stimulation is not possible with RNS, which is true but perhaps not quite to the point.

Or perhaps the authors could cite additional evidence or offer their perspective suggesting that degree of detector tuning influences efficacy, as this would be pertinent and very helpful for readers.

Manuscript No.: **COMMSMED-23-0533A**

Title: Unearthing the mechanisms of responsive neurostimulation for epilepsy

Corresponding Author: Vikram Rao

Response to Referee comments:

We are grateful for a second round of Reviewer feedback on our manuscript and for the opportunity to revise and resubmit it for consideration of publication as a Perspective in *Communications Medicine*.

Please find below our point-by-point responses to the Reviewers' concerns. The original concern is shown in *italics*. Our responses are shown in **blue**. All page numbers listed refer to pages on the manuscript version with tracked changes.

Reviewer: 1

This manuscript nicely defines the current state of RNS treatment with an emphasis on the shortfalls and lack of response that has occurred in many patients. The authors describe the current seizure focus-centric approach embraced by most centers to define appropriate targets for RNS. They then succinctly define the shortcomings of this approach and further emphasize the long-term, neuromodulatory effects of the device as opposed to its "seizure-stopping" capabilities.

The authors describe some of the foundational work supporting a more network-centric interpretation of RNS outcomes from iEEG and MEG data. They then move to the matter of volume of tissue activation, again emphasizing the ascendancy of network over focus. The use of RNS trending data to identify and define circadian and multidien variations in seizure susceptibility has received appropriately increasing emphasis, but its application to timing of stimulation affords a unique tie-in with the network-centric bent of the manuscript. Finally, the idea that seizure-associated stimulation could be less effective over time than stimulation applied during more quiescent times raises questions regarding the benefits of closed versus open loop stimulation.

It was a pleasure to read this manuscript and its description of the transition from the traditional notion of seizure foci – a notion still integral to the world of resective and ablative epilepsy surgery – to an emphasis on epileptic network dynamics – more applicable to the realm of neurostimulation and neuromodulation.

The authors have addressed this reviewer's questions, and from my perspective, this manuscript is ready for publication.

Response: We thank the Reviewer for these positive comments in support of publication of our manuscript.

Reviewer: 2

Thank you for the changes.

One minor point:

Then authors may not have fully considered the initial comment:

"It seems the closed loop aspect of RNS could readily be tested in RNS devices but never has been tested? "

Would it not be readily accomplished to set RNS detectors randomly or tuned to something other than seizure onsets, thereby testing the closed loop component? To my knowledge this has never been attempted, but would in a scientific setting be one of the first experiments/tests performed.

The authors responded that "true open loop" stimulation is not possibly with RNS, which is true but perhaps not quite to

the point.

Or perhaps the authors could cite additional evidence or offer their perspective suggesting that degree of detector tuning influences efficacy, as this would be pertinent and very helpful for readers.

Response: We apologize for not adequately addressing this insightful comment in our previous revisions. We have now added additional text and references (see below) that expand upon this important point, including the reviewer's suggestion to tune RNS detection settings to non-seizure activity (we do point out the inevitable trade-off this would have with recording seizures). Anecdotally, we have observed that clinical response sometimes depends heavily on daily stimulation rate (which is a function of detection sensitivity), but it is not clear from our experience that stimulation at seizure onset *per se* is necessary for seizure reduction.

[Page 10]: "Data from a rodent model of epilepsy suggest greater seizure reduction from closed-loop stimulation compared to open-loop stimulation with an equal energy budget¹, but testing this directly in humans is complicated by the fact that the RNS device cannot be programmed to deliver stimulation randomly or in true open-loop fashion². In principle, to determine whether stimulation at the start of seizures is necessary for seizure reduction, RNS detection settings could be tuned to patterns of neural activity that are not present at seizure onset, though this would necessarily compromise seizure quantification and other valuable diagnostic data from the device³."

References

- 1 Salam, M. T., Perez Velazquez, J. L. & Genov, R. Seizure Suppression Efficacy of Closed-Loop Versus Open-Loop Deep Brain Stimulation in a Rodent Model of Epilepsy. *IEEE Trans Neural Syst Rehabil Eng* **24**, 710-719, doi:10.1109/tnsre.2015.2498973 (2016).
- 2 Sisterson, N. D. *et al.* A Rational Approach to Understanding and Evaluating Responsive Neurostimulation. *Neuroinformatics* **18**, 365-375, doi:10.1007/s12021-019-09446-7 (2020).
- 3 Rao, V. R. Chronic electroencephalography in epilepsy with a responsive neurostimulation device: current status and future prospects. *Expert Rev Med Devices* **18**, 1093-1105, doi:10.1080/17434440.2021.1994388 (2021).

REVIEWERS' COMMENTS:

Reviewer #2 (Remarks to the Author):

Thank you for the attention to these comments.

The newly cited reference regarding open vs closed loop stimulation may not make sense in the context of this human RNS review since in the rodent example the closed-loop stimulation seems to work by stopping seizures, which the review authors already argue is not likely the way RNS works. If RNS worked by stopping seizures, then clearly tuning to seizure onset would affect outcome. So the new citation may be a bit of a red herring with a possible tendency to mislead the reader.

Also if the authors would define "and other valuable diagnostic data from the device", that would be helpful to the reader. As it is the vast majority of stimulations do not occur at the onset of the seizures.

I will leave these two points to the discretion of the authors. Thank you for your careful consideration of this important aspect of stimulation for epilepsy. I agree with Reviewer 1: "It was a pleasure to read this manuscript and its description of the transition from the traditional notion of seizure foci – a notion still integral to the world of resective and ablative epilepsy surgery – to an emphasis on epileptic network dynamics – more applicable to the realm of neurostimulation and neuromodulation."

Manuscript No.: **COMMSMED-23-0533B**

Title: Unearthing the mechanisms of responsive neurostimulation for epilepsy

Corresponding Author: Vikram Rao

Response to Referee comments:

We are grateful for a third round of Reviewer feedback on our manuscript and for the opportunity to revise and resubmit it for consideration of publication as a Perspective in *Communications Medicine*.

Please find below our point-by-point responses to the Reviewers' concerns. The original concern is shown in *italics*. Our responses are shown in blue. All page numbers listed refer to pages on the manuscript version with tracked changes.

Reviewer: 2

Thank you for the attention to these comments.

The newly cited reference regarding open vs closed loop stimulation may not make sense in the context of this human RNS review since in the rodent example the closed-loop stimulation seems to work by stopping seizures, which the review authors already argue is not likely the way RNS works. If RNS worked by stopping seizures, then clearly tuning to seizure onset would affect outcome. So the new citation may be a bit of a red herring with a possible tendency to mislead the reader.

Response: Thank you for pointing this out. We have removed the citation¹ from this section [Page 12] to avoid possibly misleading the reader. We instead include the citation in the 'Current Model' section [Page 6, reference 20] where other animal model studies are also cited.

Also if the authors would define "and other valuable diagnostic data from the device", that would be helpful to the reader. As it is the vast majority of stimulations do not occur at the onset of the seizures.

Response: For clarity, we deleted the phrase "and other valuable diagnostic data" [Page 13]. The citation² that remains in this sentence is a review article that covers applications of RNS data, including seizure quantification.

I will leave these two points to the discretion of the authors. Thank you for your careful consideration of this important aspect of stimulation for epilepsy. I agree with Reviewer 1: "It was a pleasure to read this manuscript and its description of the transition from the traditional notion of seizure foci – a notion still integral to the world of resective and ablative epilepsy surgery – to an emphasis on epileptic network dynamics – more applicable to the realm of neurostimulation and neuromodulation."

Response: We appreciate the positive comments and critical feedback, and we thank the Reviewer for their time.

References

- 1 Salam, M. T., Perez Velazquez, J. L. & Genov, R. Seizure Suppression Efficacy of Closed-Loop Versus Open-Loop Deep Brain Stimulation in a Rodent Model of Epilepsy. *IEEE Trans Neural Syst Rehabil Eng* **24**, 710-719, doi:10.1109/tnsre.2015.2498973 (2016).
- 2 Rao, V. R. Chronic electroencephalography in epilepsy with a responsive neurostimulation device: current status and future prospects. *Expert Rev Med Devices* **18**, 1093-1105, doi:10.1080/17434440.2021.1994388 (2021).